# A genome-wide association study identifies genetic loci associated with specific lobar brain volumes

Sven J. van der Lee ⓘ et al.[#]

Brain lobar volumes are heritable but genetic studies are limited. We performed genome-wide association studies of frontal, occipital, parietal and temporal lobe volumes in 16,016 individuals, and replicated our findings in 8,789 individuals. We identified six genetic loci associated with specific lobar volumes independent of intracranial volume. Two loci, associated with occipital (6q22.32) and temporal lobe volume (12q14.3), were previously reported to associate with intracranial and hippocampal volume, respectively. We identified four loci previously unknown to affect brain volumes: 3q24 for parietal lobe volume, and 1q22, 4p16.3 and 14q23.1 for occipital lobe volume. The associated variants were located in regions enriched for histone modifications (*DAAM1* and *THBS3*), or close to genes causing Mendelian brain-related diseases (*ZIC4* and *FGFRL1*). No genetic overlap between lobar volumes and neurological or psychiatric diseases was observed. Our findings reveal part of the complex genetics underlying brain development and suggest a role for regulatory regions in determining brain volumes.

F our lobes of the human brain are distinguished and several diseases can partially be attributed to lobe-specific structural changes. Functions of the frontal brain lobe include reasoning, movement, social behavior, planning, parts of speech, and problem solving[1]; functions attributed to the parietal lobe include recognition and perception of stimuli[2]; functions attributed to the temporal lobe include memory and speech[3]; and lastly, visual input is mainly processed by the occipital lobe. Brain diseases with lobe-specific abnormalities include Alzheimer's disease (in particular early onset), frontotemporal lobar degeneration[4], temporal lobe epilepsy[5], primary progressive aphasia, and cortical basilar ganglionic degeneration.

Environmental factors, such as smoking and hypertension, affect lobar brain volumes[6], but previous studies have shown that genetic differences across individuals also contribute to variability in volumetric brain measures[7,8]. The estimated heritability of brain lobar volumes is high, ranging from 26% to 84% for the frontal lobe, from 32% to 74% for the occipital lobe, from 30% to 86% for the parietal lobe, and from 55% to 88% for the temporal lobe[9–15]. In addition, genetic analyses in families suggest that the lobes are determined by independent genetic factors[15]. The observation that brain lobes are highly and differentially heritable makes them compelling targets to unravel the genetic architecture of the brain. Recent large genome-wide association studies (GWAS) have efficiently identified associations between genetic determinants and volumetric brain measures[16,17]. However, to date no genetic variants influencing brain lobar volumes have been identified. GWAS of the four lobar volumes of the brain can contribute to our understanding of brain lobe development and may provide a biological link between brain lobar volumes and brain-related traits and diseases.

To identify genetic variants of influence on lobar brain volumes, we performed GWAS of four brain lobar volumes in 16,016 individuals and replicated our findings in a sample of 8,789 individuals. We identified six loci significantly associated with specific brain lobar volumes independent of intracranial volume. With this study, we shed light on common genetic variants determining human brain volume and allow for a deepened understanding of the genetic architecture of the brain lobes.

## Results

In total, 16,016 individuals from 19 population-based or family-based cohort study and one case–control study were included in the current study. Additional information regarding the population characteristics, genotyping, and imaging methods are provided in the "Methods" section and in Supplementary Data 1–4.

**Heritability of lobar brain volumes.** Using a family-based approach, we found an age- and sex-adjusted heritability ($h^2$) for occipital lobe of 50% (95% confidence interval (CI) 38–62%), for frontal lobe of 52% (95% CI 40–64%), for temporal lobe of 59% (95% CI 49–69%), and for parietal lobe of 59% (95% CI 49–69%) (all: $p \leq 1.9 \times 10^{-19}$) (Supplementary Data 5). In comparison, the age- and sex-adjusted heritability estimate for total brain volume was 34% (95% CI 22–46%, $p = 8.8 \times 10^{-11}$).

**Novel genetic associations with brain lobar volumes.** Our multi-ethnic meta-analysis ($n = 16,016$ individuals of which 15,269 were of European ancestry) identified significant associations between genotypes and brain lobar volumes in five independent loci, even though we adjusted for intracranial volume (Figs. 1 and 2, Table 1). The quantile–quantile plots did not show high genomic inflation ($\lambda_{GC} \leq 1.05$) (Supplementary Fig. 1). Of these five loci, variants in one locus associated with temporal lobe volume, in one with parietal lobe volume, and in three with

occipital lobe volume. The variant rs146354218 (12q14.3, $p_{\text{multi-ethnic}} = 6.4 \times 10^{-10}$) associated with temporal lobe volume and rs2279829 (3q24, $p_{\text{multi-ethnic}} = 4.4 \times 10^{-10}$) associated with parietal lobe volume. Three loci associated with occipital lobe volume: index variants rs147148763 (small indel GTTGT→G, 14q23.1, $p_{\text{multi-ethnic}} = 2.9 \times 10^{-9}$), rs74921869 (4p16.3, $p_{\text{multi-ethnic}} = 6.2 \times 10^{-9}$), and rs1337736 (6q22.32, $p_{\text{multi-ethnic}} = 4.0 \times 10^{-8}$). In the European ancestry-only meta-analysis, we found a significant association with occipital lobe volume in one additional independent locus (1q22, rs12411216, $p_{\text{European ancestry-only}} = 3.9 \times 10^{-8}$). In the multi-ethnic meta-analysis, this association was below the genome-wide significance threshold ($p_{\text{multi-ethnic}} = 1.3 \times 10^{-7}$). There was no significant heterogeneity observed for any of the six significant loci (Supplementary Figs. 2–7). The sensitivity meta-analysis including only the studies using the $k$-Nearest-Neighbor (kNN) algorithm for measuring lobar volumes showed similar results compared to the studies using other methods (Supplementary Figs. 2–7). The index variants of these total six loci were common (minor allele frequency ranging from 0.13 to 0.46) and associations with volume variations were between 0.48 and 0.95 cm$^3$ per copy of the variant allele, explaining up to 0.27% of lobar volume variance per allele (Table 1). No variants were significantly associated with frontal lobe volume. All variants showing significant associations with brain lobar volumes are shown in Supplementary Data 6. Study-specific effects of all six significant loci are shown in Supplementary Figs. 2–7.

Notably, two genome-wide significant variants identified here, rs146354218 and rs2279827, were exclusively associated with the temporal lobe and parietal lobe, respectively (Supplementary Data 7). In contrast, rs147148763 and rs12411216 were not only significantly associated with occipital lobe volume but also appeared to be associated to some extent with parietal lobe volume ($p = 2.5 \times 10^{-6}$; variance explained = 0.15% and $p = 2.4 \times 10^{-5}$; variance explained = 0.11%, respectively). The other two variants also showed nominally significant associations with other lobar volumes.

**Replication.** Five out of the six index variants were available in the imputation reference panel of our replication sample ($n = 8,789$). Unfortunately, the haplotype reference consortium (HRC) reference panel does not contain insertions and deletions. Therefore, for replication we selected rs76341705, a variant that showed a comparable signal in the meta-analysis ($p_{\text{rs147148763}} = 2.9 \times 10^{-9}$, vs. $p_{\text{rs76341705}} = 4.8 \times 10^{-9}$), and in high linkage disequilibrium (LD) with the index variant ($R^2 = 0.99$). We were able to replicate all these six variants at a nominal significance level ($p$ values ranging from $3.0 \times 10^{-2}$ to $8.0 \times 10^{-7}$) with the same direction of effect as the discovery sample (Table 1, Supplementary Fig. 8).

**Variance explained in lobar volumes by common variants.** Based on the LD score regression single-nucleotide polymorphism (SNP)-based heritability analyses, common variants across the whole genome explained as much as 20.3% (95% CI 13.2–27.4%) of the variance in occipital lobe volume, 19.6% (95% CI 12.3–26.9%) of frontal lobe volume, 17.5% (95% CI 10.7–24.3%) of temporal lobe volume, and 17.9% (95% CI 11.7–24.1%) of parietal lobe volume (Supplementary Data 8). Common genetic variants account for 30–41% of the total heritability of brain lobar volumes (Supplementary Data 8).

**Genetic overlap with other brain volumes and related diseases.** Although no top variant was significantly associated with the volume of multiple lobes, nominally significant correlation ($r_g$)

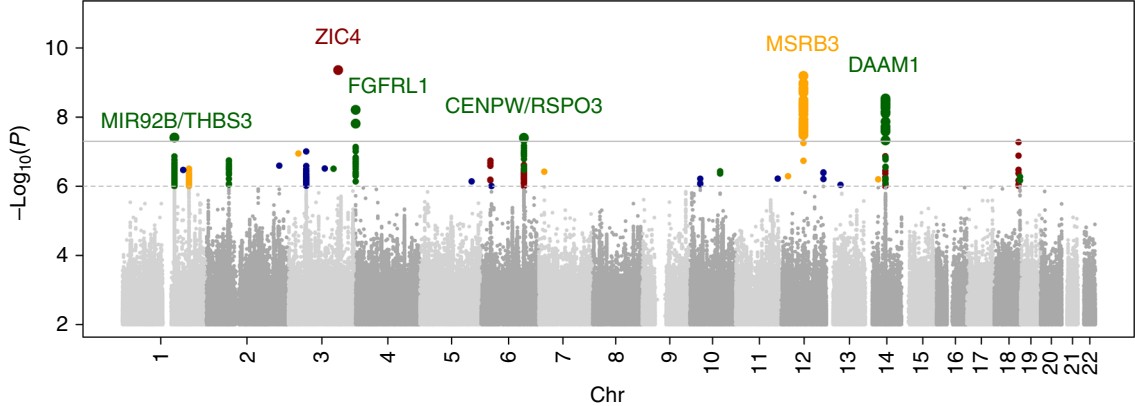

**Fig. 1** Common genetic variants associated with frontal, parietal, temporal and occipital lobe volume. 'Manhattan' plot displaying the association $p$ value for each tested single-nucleotide polymorphism (SNP) (displayed as $-\log_{10}$ of the $p$ value). Genome-wide significance threshold is shown with a line at $p = 5 \times 10^{-8}$ (solid black line) and also the suggestive threshold at $p = 1 \times 10^{-5}$ (dashed line). Dots represent SNPs, results of the four lobes are shown in a single figure, and the nearby gene is labeled. Above the suggestive threshold SNPs are colored by the associated lobe: yellow = temporal lobe, green = occipital lobe, red = parietal lobe, blue = frontal lobe

**Fig. 2** Regional view of the genome-wide significant loci. For each panel, zoomed in Manhattan plots (±kb from top single-nucleotide polymorphism (SNP)) are shown with gene models below (GENCODE version 19). Plots are zoomed in to highlight the genomic region that contains the index SNP and SNPs in linkage disequilibrium with the index SNP ($R^2 > 0.8$). Each plot was made using the LocusTrack software (http://gump.qimr.edu.au/general/gabrieC/LocusTrack/)

**Table 1 Genetic variants at six loci significantly associated with lobar brain volumes**

| Lobe | Annotation (nearby) | Gene | Rs-ID | Chr | Position | A1 | A2 | European ancestry meta-analysis | | | | Multi-ethnic meta-analysis | | | | Replication sample (n = 8,789) | | | | | |
|---|---|---|---|---|---|---|---|---|---|---|---|---|---|---|---|---|---|---|---|---|---|
| | | | | | | | | Frq | Effect | SE | p | Frq | Effect | SE | p | N | R² (%) | Frq | Effect | SE | p |
| Parietal | UTR3 | ZIC4[16] | rs2279829 | 3 | 147106319 | T | C | 0.21 | −0.92 | 0.16 | $5.3 \times 10^{-9}$ | 0.21 | −0.95 | 0.15 | $4.4 \times 10^{-10}$ | 16,015 | 0.24 | 0.22 | −0.63 | 0.18 | $6.0 \times 10^{-4}$ |
| Temporal | Intronic | MSRB3 | rs146354218 | 12 | 65793942 | A | G | 0.37 | 0.69 | 0.11 | $1.2 \times 10^{-9}$ | 0.37 | 0.69 | 0.11 | $6.4 \times 10^{-10}$ | 15,789 | 0.24 | 0.37 | 0.54 | 0.14 | $6.4 \times 10^{-5}$ |
| Occipital | Intergenic | DAAM1 (dist = 24302) | rs147148763[a] | 14 | 59631075 | G | GTTGT | 0.13 | −0.83 | 0.14 | $6.6 \times 10^{-9}$ | 0.13 | −0.85 | 0.14 | $2.9 \times 10^{-9}$ | 15,220 | 0.23 | 0.12 | −0.80 | 0.12 | $3.1 \times 10^{-11}$ |
| Occipital | Intronic | FGFRL1 | rs74921869 | 4 | 1013382 | A | G | 0.2 | −0.84 | 0.14 | $5.9 \times 10^{-9}$ | 0.2 | −0.82 | 0.14 | $6.2 \times 10^{-9}$ | 12,424 | 0.27 | 0.19 | −0.22 | 0.10 | $3.0 \times 10^{-2}$ |
| Occipital | Intergenic | CENPW (dist = 175626) | rs1337736 | 6 | 126845380 | A | G | 0.23 | −0.64 | 0.12 | $8.1 \times 10^{-8}$ | 0.23 | −0.64 | 0.12 | $4.0 \times 10^{-8}$ | 16,016 | 0.19 | 0.23 | −0.46 | 0.09 | $8.0 \times 10^{-7}$ |
| Occipital | Upstream | MIR92B/THBS3 | rs1241216 | 1 | 155164480 | A | C | 0.46 | −0.52 | 0.10 | $3.9 \times 10^{-8}$ | 0.46 | −0.49 | 0.09 | $1.4 \times 10^{-7}$ | 16,016 | 0.17 | 0.45 | −0.19 | 0.08 | $1.6 \times 10^{-2}$ |

The allele frequency (frq) and effect size are given for A1. Effect sizes are given in units of $cm^3$ per effect allele. Results are provided for the discovery samples and the meta-analysis of all European ancestry and the multi-ethnic meta-analysis. The variance explained gives the percentage variance explained of a SNP[16]

A1 effect allele, A2 reference allele, Chr chromosome, Frq effect allele frequency, N number of individuals with genetic variant, p p value, Rs-ID reference SNP cluster ID, SE standard error

[a]In the replication sample, another significant variant in high LD ($R^2 = 0.99$) with this variant was used (rs76341705, p value = $4.8 \times 10^{-9}$)

Abbreviations: effect allele (A1), reference allele (A2), chromosome (Chr), effect allele frequency (Frq), number of individuals with genetic variant (N), p-value (P), reference SNP cluster ID (Rs-ID), standard error (SE)

was observed between genetic components of the parietal and temporal lobe ($r_g = 0.35$, $p = 1.5 \times 10^{-3}$), although this did not withstand correction for multiple testing (Fig. 3, Supplementary Data 9). Some suggestive correlation was observed between temporal and frontal lobe volume with genetic determinants of subcortical volumes; however, none survived multiple testing adjustments. When studying brain diseases, only occipital lobe volume showed a suggestive genetic correlation with Parkinson's disease ($r_g = 0.18$, $p = 0.03$). No significant genetic correlation was observed with any of the other tested neurological or psychiatric traits.

## Discussion

In our genome-wide association study of in up to 16,016 individuals, we identified 6 independent loci where variants had significant associations with brain lobar volumes, independent of intracranial volume. We were able to replicate these findings in a sample of 8,789 individuals. Four out of the six identified loci have not been linked to brain volume measures before; the other two loci are located in regions previously associated with brain volume measures (12q14.3 with hippocampal volume and 6q22.32 with intracranial volume). These new loci provide intriguing new insights into the genetics underlying brain lobar volumes.

We estimated that, after adjusting for intracranial volume, 17.5–20.3% of the variance in lobar volumes could be explained by common genetic variation. This forms 30–40% of the total heritability we estimated, suggesting a major contribution of common genetic variation in brain development. More genetic variants associated with brain volume may be discovered by increasing the sample sizes of genetic studies. An interesting observation is that we were able to replicate our findings using only gray matter volumes of each lobe, while the discovery studied the sum of gray and white matter. This difference might explain that not all loci replicated as strong as others and that differential effects might exist on gray and white matter volume effects of the genetic variants. Future studies will have to elucidate the biological mechanisms of the discovered associations.

Interestingly, the majority of the identified loci contained variants associated with occipital lobe volume, whereas the other brain lobes have more often been linked to disease outcomes. Yet, the heritability estimates for the occipital lobe do not exceed the heritability estimates of the other brain lobes. One possible explanation for this finding is the smaller volume of the occipital lobe compared to the other lobes, making it a more specific region —also in terms of the genetic architecture. It may also be explained by a less polygenic nature of the occipital lobe compared to the other lobes, allowing one to identify stronger associations for a single genetic variant.

Regarding the identified genome-wide significant loci, two identified loci have been previously associated with brain volume measurements. The locus 12q14.3 associated with temporal lobe volume in our study and was previously associated with hippocampal volume[17]. Our index variant rs146354218 ($p = 6.4 \times 10^{-10}$) is an intronic variant in the *MSRB3* gene and lies 39 kilobases (kb) from the previously published rs61921502 variant associated with hippocampal volume[18]; however, the LD ($R^2 = 0.1$, D' = 1, $p < 0.0001$)[19] is low. This previously published variant also showed some evidence of association with total temporal lobe volume (effect = 0.57 $cm^3$, $p = 6.5 \times 10^{-4}$). Thus the 12q14.3 locus not only influences hippocampal volume but also seems to have a more generalized effect on the temporal lobe volume as would be expected by the genetic correlation between temporal lobe volume and hippocampal volume. The signal (rs1337736, $p = 4.0 \times 10^{-8}$) at 6q22.32 near to the gene *CENPW* (Centromere

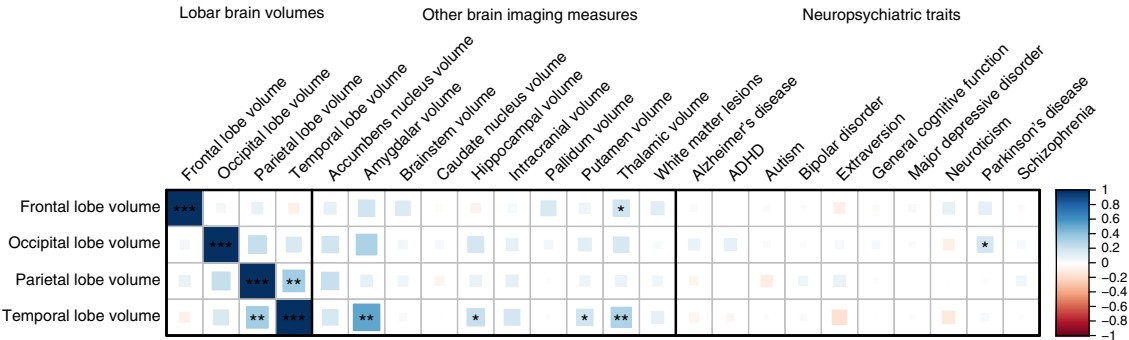

**Fig. 3** Genetic correlation between lobar brain volumes and other brain imaging measures and neuropsychiatric traits. Heatmap showing the genetic correlations estimates ($r_g$) as calculated by linkage disequilibrium score regression. Larger blocks and darker colors present stronger correlations, with blue and red indicating positive and negative correlations, respectively. The strength of the significance levels are indicated by asterisks: *$p < 0.05$; **$p < 1.3 \times 10^{-2}$ (0.05/4), adjusted for the lobe count; ***$p < 5.8 \times 10^{-4}$ (0.05/86), adjusted for the number of correlations tested

Protein W) is associated with occipital volume. This signal overlapped previously associated signals with intracranial volume[16,17] and is further implicated in bone mineral density[20], height[21], waist-hip ratio[22], and infant length[23]. The index variant associated with intracranial volume (rs11759026) and our top variant are in linkage equilibrium ($R^2 = 0.07$, $D' = 1$, $p = 0.0002$)[19]. We also found suggestive associations between rs11759026 and both frontal ($-1.0$ cm$^3$, $p = 6.3 \times 10^{-5}$) and occipital lobar volume ($-0.31$ cm$^3$, $p = 6.6 \times 10^{-3}$). Each locus was located in regions that are under epigenetic regulation in brain tissue (Supplementary Data 6) or close to genes or genomic loci associated with Mendelian brain-related diseases. The variant rs2279829 (3q24) is located in the 3'-untranslated region (UTR) of the Zic Family Member 4 (*ZIC4*) gene and close to the related *ZIC1* gene. This variant localizes within enhancer sites in predominantly neurological cell types, among which the brain germinal matrix (Supplementary Data 6) and both *ZIC4* and *ZIC1* are expressed throughout the brain (Supplementary Fig. 9). Heterozygous deletions of *ZIC1* and *ZIC4* cause Dandy–Walker malformation[24]. Children with this malformation have no vermis, the part connecting the two cerebellar hemispheres[24]. Gain-of-function mutations in *ZIC1* lead to coronal craniosynostosis and learning disability[25]. Variant rs147148763 was located 24 kb from the disheveled-associated activator of morphogenesis 1 (*DAAM1*). There is evidence for the most significant variants to localize within enhancer sites, as well as DNA-hypersensitivity sites in brain tissues. Also, genome-wide significant SNPs in the locus are expression quantitative trait loci (eQTLs) of *DAAM1* in blood (Supplementary Data 6). Daam1 is a formin protein that has been linked to actin dynamics[26], is regulated by RhoA[27], and is expressed in the shafts of dendrites[28]. Expression patterns in brain development of animals further suggest a role in neuronal cell differentiation and movement[29]. Variant rs4647940 is located in the 3'-UTR of fibroblast growth factor receptor (*FGFRL1*) and is in LD with a missense variant in Alpha-L-iduronidase (*IDUA*) (rs3755955, $R^2 = 0.87$) that was previously associated with bone mineral density ($p = 5.0 \times 10^{-15}$). Deletion of the 4p16.3 locus causes Wolf–Hirschhorn syndrome, a neurodevelopmental disorder characterized by mental retardation, craniofacial malformation, and defects in skeletal and heart development. Variant rs12411216 is located in an intron of *MIR92B* and *THBS3*, but the signal peak in this locus covers >20 genes. Promotor histone marks overlap the variant and it is an eQTL for multiple genes, both in a multitude of different tissues among which brain tissues (Supplementary Data 6). In summary, these findings link genes that cause Mendelian syndromes affecting cranial skeletal

malformations, brain malformations, and intelligence with brain lobe volume in healthy individuals. One other interesting variant in tight LD with our index variant (rs4072037, $R^2 = 0.94$) is a missense SNP in the *MUC1* gene that decreased levels of blood magnesium concentrations[30]. It is not clear how decreased magnesium levels are involved in decreased occipital brain volume, but it is an interesting avenue to explore as magnesium is known to be important for neural transmissions[31] and magnesium infusions have anti-convulsive effects and is still used to prevent convulsions in pre-eclampsia[32].

Using genetic correlation analysis, we did not find a strong significant genetic correlation between most of the brain lobes, which suggests that the genetic basis of the brain lobes is largely independent[15]. We also did not find significant genetic overlap between lobar brain volumes and neurological and psychiatric disease outcomes. The most significant genetic correlation with brain lobar volume and diseases we observed was between occipital lobe volume and Parkinson's disease ($r_g = 0.18$, $p = 0.03$). However, this finding was not significant after multiple testing correction, leading us to report this finding with caution. The absence of significant genetic correlations between other brain lobes and clinical diseases could be due to true absence of a genetic overlap. However, other explanations can be put forward. First, it could also mean that our lobar volume GWAS and those for other diseases were still too underpowered to show significant genetic correlations. Second, the anatomical boundaries for the different lobes can be quite arbitrary and do not necessarily have to coincide with underlying gene function or biological processes leading to neurological or psychiatric disorders.

There are several limitations to our study. First, we have accepted differences in analytical methods of the magnetic resonance imaging (MRI) scans to allow for the largest sample size to be studied. This might have resulted in different effects over the studies. However, we did not observe significant heterogeneity after correcting for multiple testing for the six loci. In addition, a sensitivity analysis showed similar effects of the genome-wide significant variants for the studies using the kNN algorithm in comparison to the other studies. False negative findings due to differences in analytical methods of the MRI scans cannot be excluded. Second, a limitation of our study is that a different reference panel for imputation was used for the discovery and replication sample. This is due to the historic limited availability of the HRC reference panel at the initiation of this study. As the variants were well imputed ($R^2 > 0.5$) in all studies, this is not expected to have influenced the results, although it is possible that additional variants may be discovered if the larger HRC reference

panel would be used in future studies. Last, for the UK Biobank only gray matter parcellations were available to us. Despite this limitation, we were able to replicate our findings. This suggests that the identified variants have an effect on gray as well as white matter volumes.

In summary, brain lobar volumes are differentially heritable traits, which can in large part be explained by common genetic variation. We identified six loci where genotypes are associated with specific brain lobes, four of which have not been implicated in brain morphology before. These loci are compelling targets for functional research to identify the biology behind their genetic signals.

## Methods

**Study population**. The study sample consisted of dementia- and stroke-free individuals with quantitative brain MRI and genome-wide genotypes from 19 population- and family-based cohort studies participating in the Cohorts of Heart and Aging Research in Genomic Epidemiology consortium and the case–control Alzheimer's Disease Neuroimaging Initiative study. In total, 16,016 participants were included, 15,269 participants of European ancestry, 405 African Americans, 211 Chinese, and 131 Malay. We attempted to replicate our findings in 8,789 European ancestry individuals from the UK Biobank, an ongoing prospective population-based cohort study located in the United Kingdom. Descriptive statistics of all populations are provided in Supplementary Data 1. DNA from whole blood was extracted and genome-wide genotyping was performed using a range of commercially available genotyping arrays. Genotype imputations were performed in each discovery cohort using 1000 Genomes version 1[33] as reference and using the Haplotype Reference Consortium (HRC) version 1.1 in the replication cohort (Supplementary Data 2).

**MRI methods**. Three-dimensional T1-weighted brain MRI data were acquired by each cohort (Supplementary Data 1). Cohorts in the discovery sample segmented the T1-weighted images into supra-tentorial gray matter, white matter, and cerebrospinal fluid. The methods of image segmentation varied across study cohorts (Supplementary Methods). However, the majority used a previously described kNN algorithm, which was trained on six manually labeled atlases[34], or in-house image-processing pipelines. In each study, MRI scans were performed and processed with automated protocols, without reference to clinical or genetic information. We studied the total volume (sum of white and gray matter and the left and right hemisphere) of the frontal, parietal, temporal, and occipital brain lobes, adjusted for intracranial volume. Descriptive information of the lobar volumes across the different studies is provided in Supplementary Data 3. Differences in average brain lobar volumes were accepted as differences in MRI acquisition, processing, segmentation, and demographics, which exist over cohorts. As a replication, we used the released volume measurements of 8,789 UK Biobank participants, extracted using the FreeSurfer software version 6.0, which obtains lobar volumes by adding up regions of interest volumes[35,36]. As only FreeSurfer gray matter volumes were available for this study sample, the replication sample volumes were smaller than the volumes in the discovery sample (Supplementary Data 3).

**Estimation of heritability**. The heritability of lobar brain volumes was estimated using family structure in the Framingham Heart Study ($n = 2080$), which constitutes a community-based cohort of non-demented individuals without evidence of significant brain injury (e.g., stroke or multiple sclerosis). In total, 619 extended families with a family size of $3.6 \pm 6.6$ individuals were included in the analyses. These families consisted of the following pairs of relatives: 316 parent–offspring, 1135 sibling, 340 avuncular, 1772 first cousin, and 826 second cousin pairs. We calculated additive genetic heritability without shared environmental effects ($C$) using a variance-components analysis under an AE model in SOLAR[37], adjusted for age, age$^2$, and sex.

**GWAS of lobar volumes**. Associations of imputed genotype dosages with lobar volumes were examined using linear regression analyses under an additive model. Associations were adjusted for age, age$^2$, sex, the first four principal components to account for possible confounding due to population stratification, and study-specific covariates. Linear mixed models with estimated kinships were used for association analyses in cohorts with related samples. Details on the analysis methods used in each cohort are provided in Supplementary Data 1 and 2. Post-GWAS quality control (QC) was conducted using EasyQC[38] and filtering. Genetic variants with a low imputation quality ($R^2 < 0.5$), a minor allele count <10, and allelic or locational mismatching of SNPs with the reference panel were removed prior to the meta-analyses. The number of variants after filtering and the genomic inflation per study are provided in Supplementary Data 4. After QC, summary statistics were adjusted by the genomic control method in each of the participating cohorts[39]. We then performed two inverse-variance weighted fixed-effect meta-analyses in METAL[39]. First, we meta-analyzed all participants of European

ancestry, then performed a multi-ethnic meta-analysis including African Americans ($n = 405$), Chinese ($n = 211$), and Malay ($n = 131$). After meta-analyses, genetic variants with a total sample size of <5000 were excluded. We performed conditional analysis on the index variants to determine whether there were multiple independent genome-wide significant variants in a locus using the Genome-Wide Complex Trait Analysis (GCTA) software (--cojo, --p-cojo)[40,41]. Genotypes in the Rotterdam-study (all 6291 individuals of the baseline cohort who were genotyped) were used as reference for this analysis. For loci with a genome-wide significant association ($p < 5 \times 10^{-8}$), we tested for heterogeneity using the $I^2$ statistic[39]. In a sensitivity meta-analysis, we tested whether the studies using the kNN algorithm had similar effects of the genome-wide significant variants for the studies using the kNN-algorithm in comparison to the other studies. We also searched for candidate genes in the loci using publically available databases for differential expression of the SNPs (eQTL database in GTEx)[42] and HaploReg, an online tool that summarizes the ENCODE database for epigenetic markings and proteins binding to DNA[43].

**Variance explained by common variants and genetic correlations**. The variance explained by all SNPs, or SNP-based heritability, was calculated from summary statistics using LD score regression[44]. The percentage of variance explained by all SNPs was determined based on meta-analysis results using the LD Hub[45]. We used the same LD score regression[44] to quantify the amount of genetic correlation between the four brain lobes and other brain-related traits and diseases, using summary statistics for meta-analyses of genetic studies of subcortical structures[16], intracranial volume[17], white matter hyperintensities[46], general cognitive ability[47], neuroticism[48], schizophrenia[49], attention-deficit/hyperactivity disorder[50], autism[51], major depressive disorder[52], bipolar disorder,[49] Parkinson's disease[53], and Alzheimer's disease[54].

**Ethical compliance**. All participants, or their parents or guardians in the case of minors, provided written informed consent for study participation, the use of brain MRI data, and the use of their DNA for genetic research. Approval for the individual studies was obtained by the relevant local ethical committees and institutional review boards.

**Statistics and reproducibility**. Software used for the data analysis of this study: EasyQC (www.genepi-regensburg.de/easyqc), FreeSurfer (https://surfer.nmr.mgh.harvard.edu/), GCTA (http://cnsgenomics.com/software/gcta/), GenABEL (http://www.genabel.org), GTeX (https://gtexportal.org/home/), HaploReg (https://www.encodeproject.org/software/haploreg/), HASE (https://github.com/roshchupkin/hase), LD-hub (http://ldsc.broadinstitute.org/ldhub/), LD score regression (https://github.com/bulik/ldsc), mach2qtl (https://www.nitrc.org/projects/mach2qtl/), METAL (http://csg.sph.umich.edu/abecasis/metal/), Perl (https://www.perl.org/), PLINK (https://www.cog-genomics.org/plink2), R (https://www.r-project.org/), SNPTEST (https://mathgen.stats.ox.ac.uk/genetics_software/snptest/snptest.html), and SOLAR (http://www.sfbr.org).

**Reporting summary**. Further information on research design is available in the Nature Research Reporting Summary linked to this article.

## Data availability

The genome-wide summary statistics that support the findings of this study will be made available via the NHGRI-EBI GWAS Catalog website (https://www.ebi.ac.uk/gwas/downloads/summarystatistics) upon publication. Quantitative brain MRI and genotype data are available from the corresponding authors H.H.H.A. and C.D.C. upon reasonable request.

## Code availability

No previously unreported custom computer code or mathematical algorithm was used to generate results central to the conclusions. The code is available upon request from the corresponding authors H.H.H.A. and C.D.C.

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

## Acknowledgements

Data used in preparing this article were obtained from the Alzheimer's Disease Neuroimaging Initiative (ADNI) database (adni.loni.usc.edu). As such, many investigators within the ADNI contributed to the design and implementation of ADNI and/or provided data but did not participate in analysis or writing of this report. A complete listing of ADNI investigators can be found at: http://adni.loni.usc.edu/wp-content/uploads/how_to_apply/ADNI_Acknowledgement_List.pdf. Full acknowledgements are provided in the Supplementary Information.

## Author contributions

(Co)wrote draft: S.J.v.d.L., M.J.K., N.A., H.H.H.A., C.D. Meta-analysis: S.J.v.d.L. Analyzed data: G.C., C.S., A.V.S., E.H., J.C.B., D.P.H., S.H., E.B.v.dA., M. Bernard, L.R.Y., F.C., J.W.C., Y.S., S.L., L.Y., G.V.R., N.J., Q.Y., R.A.M., S.B., C.Y.C., P.M., W.Z., P.A.N., B.M.P., Z.P., P.M.T. Critical revision of manuscript: G.C., C.S., A.V.S., E.H., J.C.B., D.P.H., E.B.v.A., M. Bernard, L.R.Y., N.A., J.W.C., Y.S., O.L.L., S.L., J.v.dG., L.Y., T.P., G.V.R., P.A., N.J., Q.Y., R.A.M., S.B., B.M., C.Y.C., P.M., W.J.N., M. Beekman, W.Z., P.A.N., M.W.V., H.S., Z.P., D.M.B., P.L.D.J., P.M.T., C.M.v.D., R.S., W.T.L., M.A.I., S.S. Contributed to sample collection: J.C.B., S.H., K.A., T.B.H., O.L.L., J.v.dG., T.P., K.D.T., B.M., C.Y.C., K.R., D.v.H., T.Y.W., W.J.N., A.S.B., M. Beekman, P.A.N., C.C., L.L., B.M.P., M.K.I., M.W.V., H.S., Z.P., D.M.B., P.L.D.J., C.M.v.D., D.A.B., P.E.S., R.S., W.T.L., M.A.I., S.S., S.D., V.G. Secured funding: K.A., T.B.H., O.L.L., P.A., K.D.T., B.M.,

K.R., D.v.H., T.Y.W., W.J.N., A.S.B., C.C., L.L., B.M.P., M.K.I., H.S., D.M.B., P.L.d.J., C.M.v.D., D.A.B., P.E.S., R.S., W.T.L., M.A.I., S.S., S.D., V.G. Study supervision: H.H.H.A., C.D.

## Additional information

**Competing interests:** The authors declare no competing interests.

Sven J. van der Lee[1], Maria J. Knol[1], Ganesh Chauhan[2,3], Claudia L. Satizabal[4,5], Albert Vernon Smith[6,7], Edith Hofer[8,9], Joshua C. Bis[10], Derrek P. Hibar[11], Saima Hilal[1,12,13,14], Erik B. van den Akker[15,16,17], Konstantinos Arfanakis[18,19], Manon Bernard[20], Lisa R. Yanek[21], Najaf Amin[1], Fabrice Crivello[22], Josh W. Cheung[11], Tamara B. Harris[23], Yasaman Saba[24], Oscar L. Lopez[25], Shuo Li[26], Jeroen van der Grond[27], Lei Yu[19], Tomas Paus[28,29], Gennady V. Roshchupkin[1,14,30], Philippe Amouyel[31], Neda Jahanshad[11], Kent D. Taylor[32], Qiong Yang[26], Rasika A. Mathias[21], Stefan Boehringer[17], Bernard Mazoyer[22], Ken Rice[33], Ching Yu Cheng[34], Pauline Maillard[35], Diana van Heemst[36], Tien Yin Wong[34], Wiro J. Niessen[30,37], Alexa S. Beiser[5,26], Marian Beekman[15], Wanting Zhao[34], Paul A. Nyquist[38], Christopher Chen[12,13], Lenore J. Launer[23], Bruce M. Psaty[10,39,40,41], M. Kamran Ikram[1,42], Meike W. Vernooij[1,14], Helena Schmidt[24], Zdenka Pausova[20,43], Diane M. Becker[21], Philip L. De Jager[44,45], Paul M. Thompson[11], Cornelia M. van Duijn[1], David A. Bennett[19], P. Eline Slagboom[15], Reinhold Schmidt[8], W.T. Longstreth[39,46], M. Arfan Ikram[1], Sudha Seshadri[4,5], Stéphanie Debette[2,47], Vilmundur Gudnason[6,7], Hieab H.H. Adams[1,14,48] & Charles DeCarli[49]

[1]Department of Epidemiology, Erasmus MC University Medical Center, Rotterdam 3015CN, the Netherlands. [2]University of Bordeaux, Bordeaux Population Health Research Center, INSERM UMR 1219, 33000 Bordeaux, France. [3]Centre for Brain Research, Indian Institute of Science, Bangalore 560012, India. [4]The Glenn Biggs Institute for Alzheimer's and Neurodegenerative Diseases, UT Health San Antonio, San Antonio, TX 78229, USA. [5]Boston University School of Medicine and the Framingham Heart Study, Boston, MA 02118, USA. [6]Icelandic Heart Association, 201 Kopavogur, Iceland. [7]Faculty of Medicine, University of Iceland, 101 Reykjavik, Iceland. [8]Clinical Division of Neurogeriatrics, Department of Neurology, Medical University of Graz, Graz 8036, Austria. [9]Institute for Medical Informatics, Statistics and Documentation, Medical University of Graz, Graz 8036, Austria. [10]Cardiovascular Health Research Unit, Department of Medicine, University of Washington, Seattle, WA 98101, USA. [11]Imaging Genetics Center, Mark and Mary Stevens Neuroimaging & Informatics Institute, Keck School of Medicine of the University of Southern California, Los Angeles, CA 90292, USA. [12]Department of Pharmacology, National University of Singapore, Singapore 117600, Singapore. [13]Memory, Aging and Cognition Center, National University Health System, Singapore 119228, Singapore. [14]Department of Radiology and Nuclear Medicine, Erasmus MC University Medical Center, Rotterdam 3015CN, the Netherlands. [15]Department of Biomedical Data Sciences, Section of Molecular Epidemiology, Leiden University Medical Center, Leiden 2333ZA, the Netherlands. [16]Pattern Recognition & Bioinformatics, Delft University of Technology, Delft 2628XE, the Netherlands. [17]Department of Biomedical Data Sciences, Statistical Genetics, Leiden University Medical Center, Leiden 2333ZA, the Netherlands. [18]Department of Biomedical Engineering, Illinois Institute of Technology, Chicago, IL 60616, USA. [19]Rush Alzheimer's Disease Center, Rush University Medical Center, Chicago, IL 60612, USA. [20]The Hospital for Sick Children, University of Toronto, Toronto M5G 1X8 ON, Canada. [21]GeneSTAR Research Program, Department of Medicine, Johns Hopkins School of Medicine, Baltimore, MD 21205, USA. [22]Neurofunctional Imaging Group - Neurodegenerative Diseases Institute, UMR 5293, Team 5 - CEA - CNRS - Bordeaux University, Bordeaux 33076, France. [23]Laboratory of Epidemiology and Population Sciences, National Institute on Aging, Intramural Research Program, National Institutes of Health, Bethesda, MD 20892, USA. [24]Research Unit-Genetic Epidemiology, Gottfried Schatz Research Centre for Cell Signaling, Metabolism and Aging, Molecular Biology and Biochemistry, Medical University of Graz, 8010 Graz, Austria. [25]Department of Neurology, University of Pittsburgh, Pittsburgh, PA 15260, USA. [26]Department of Biostatistics, School of Public Health, Boston University, Boston, MA 02118, USA. [27]Department of Radiology, Leiden University Medical Center, Leiden 2333ZA, the Netherlands. [28]Bloorview Research Institute, Holland Bloorview Kids Rehabilitation Hospital, Toronto M4G 1R8, Canada. [29]Departments of Psychology and Psychiatry, University of Toronto, Toronto M5S 1A1, Canada. [30]Department of Medical Informatics, Erasmus MC University Medical Center, Rotterdam 3015CN, the Netherlands. [31]Univ. Lille, Inserm, Centre Hosp. Univ Lille, Institut Pasteur de Lille, LabEx DISTALZ-UMR1167 - RID-AGE - Risk factors and molecular determinants of aging-related, 59000 Lille, France. [32]Institute for Translational Genomics and Population Sciences, Department of Pediatrics at LABioMed-Harbor-UCLA Medical Center, Torrance, CA 90502, USA. [33]Department of Biostatistics, University of Washington, Seattle, WA 98195, USA. [34]Singapore Eye Research Institute, Singapore National Eye Center, Singapore 169857, Singapore. [35]Imaging of Dementia and Aging (IDeA) Laboratory, University of California-Davis,

Davis, CA 95817, USA. [36]Department of Gerontology and Geriatrics, Leiden University Medical Center, Leiden 2333ZA, the Netherlands. [37]Faculty of Applied Sciences, Delft University of Technology, Delft 2629HZ, the Netherlands. [38]Department of Neurology, Johns Hopkins School of Medicine, Baltimore, MD 21205, USA. [39]Department of Epidemiology, University of Washington, Seattle, WA 98195, USA. [40]Department of Health Services, University of Washington, Seattle, WA 98195, USA. [41]Kaiser Permanente Washington Health Research Institute, Seattle, WA 98101, USA. [42]Department of Neurology, Erasmus MC University Medical Center, Rotterdam 3015CN, the Netherlands. [43]Departments of Physiology and Nutritional Sciences, The Hospital for Sick Children, University of Toronto, Toronto M5G 1X8, Canada. [44]Center for Translational and Computational Neuroimmunology, Columbia University Medical Center, New York, NY 10032, USA. [45]Program in Medical and Population Genetics, Broad Institute, Cambridge, MA 02142, USA. [46]Department of Neurology, University of Washington, Seattle, WA 98195, USA. [47]Department of Neurology, University Hospital of Bordeaux, Bordeaux 33000, France. [48]Department of Clinical Genetics, Erasmus MC University Medical Center, Rotterdam, The Netherlands. [49]Department of Neurology and Center for Neuroscience, University of California at Davis, Davis, CA 95817, USA

