## [Peer Review File · Communications Biology]

Editorial Note: This manuscript has been previously reviewed at another journal. This document only contains reviewer comments and rebuttal letters for versions considered at *Communications Biology*.

REVIEWERS' COMMENTS:

Reviewer #1 (Remarks to the Author):

COMMSBIO-19-0279-T

Novel genetic loci associated with brain lobar volumes

The authors have addressed most of my questions but there are some additional minor issues.

* Does FreeSurfer directly output lobar volume estimates? Or is the lobar volume calculated as the sum of multiple regional volumes?

* Page 8 - "LD score regression implemented in GCTA" hasn't been corrected.

* Same paragraph - LD hub is designed to estimate genetic correlation between traits, not for estimating variance explained by top SNPs. The description is confusing.

* Page 9 - "The effect estimates were comThe index variants..." Please correct the typo.

* Page 13 - "we did not observe no significant heterogeneity..."

Reviewer #2 (Remarks to the Author):

the authors have adequately responded to the issues raised.

Please review the editorial comments and requests below and confirm the changes made in the manuscript in the rightmost column. There may be additional requests within the decision letter email or other attachments.

Response to the reviewers' comments	
Does FreeSurfer directly output lobar volume estimates? Or is the lobar volume calculated as the sum of multiple regional volumes?	FreeSurfer does not directly output lobar volume estimates, but this is a sum of multiple regional volumes. More information is provided in the Supplementary Methods sections. We also added some additional information to the Methods section: 'extracted using FreeSurfer software version 6.0, which obtains lobar volumes by adding up regions of interest volumes'
Page 8 - "LD score regression implemented in GCTA" hasn't been corrected.	Apologies, we have now corrected this and removed 'implemented in GCTA' from the sentence.
Same paragraph - LD hub is designed to estimate genetic correlation between traits, not for estimating variance explained by top SNPs. The description is confusing.	LD score regression allows to calculate both genetic correlations with other traits as well as the SNP-based heritability of a trait, which has also been implemented in the LD Hub. However, the reviewer is correct that this is not only based on genome-wide significant variants but on all GWAS variants. We have therefore changed the sentence as following: 'The percentage of variance explained by all SNPs was determined based on meta-analysis results using the LD-hub.'
Page 9 - "The effect estimates were comThe index variants..." Please correct the typo.	We have now removed the first part of the sentence ('The effect estimates were com').
Page 13 - "we did not observe no significant heterogeneity..."	This has been changed: 'we did not observe significant heterogeneity'.